# The Current Role of Dexmedetomidine as Neuroprotective Agent: An Updated Review

**DOI:** 10.3390/brainsci11070846

**Published:** 2021-06-25

**Authors:** Zaara Liaquat, Xiaoying Xu, Prince Last Mudenda Zilundu, Rao Fu, Lihua Zhou

**Affiliations:** 1Department of Anatomy, School of Medicine, Sun Yat-sen University, Shenzhen 518100, China; liaquat@mail2.sysu.edu.cn (Z.L.); zhoulih@mail.sysu.edu.cn (L.Z.); 2Department of Anatomy, Zhongshan School of Medicine, Sun Yat-sen University, Guangzhou 510080, China; xuxy43@mail2.sysu.edu.cn (X.X.); prince.zilundu@nust.ac.zw (P.L.M.Z.)

**Keywords:** apoptosis, caspase 8, dexmedetomidine, excitatory amino acid transporter 3, excitotoxic neurotransmitter, focal adhesion kinase, necrosis, nerve injury, neuroprotection, neuroinflammation

## Abstract

Dexmedetomidine, selective α2-adrenergic agonist dexmedetomidine, has been widely used clinically for sedation and anesthesia. The role of dexmedetomidine has been an interesting topic of neonatological and anesthetic research since a series of advantages of dexmedetomidine, such as enhancing recovery from surgery, reducing opioid prescription, decreasing sympathetic tone, inhibiting inflammatory reactions, and protecting organs, were reported. Particularly, an increasing number of animal studies have demonstrated that dexmedetomidine ameliorates the neurological outcomes associated with various brain and spinal cord injuries. In addition, a growing number of clinical trials have reported the efficacy of dexmedetomidine for decreasing the rates of postoperative neurological dysfunction, such as delirium and stroke, which strongly highlights the possibility of dexmedetomidine functioning as a neuroprotective agent for future clinical use. Mechanism studies have linked dexmedetomidine’s neuroprotective properties with its modulation of neuroinflammation, apoptosis, oxidative stress, and synaptic plasticity via the α2-adrenergic receptor, dependently or independently. By reviewing recent advances and preclinical and clinical evidence on the neuroprotective effects of dexmedetomidine, we hope to provide a complete understanding of the above mechanism and provide insights into the potential efficacy of this agent in clinical use for patients.

## 1. Introduction

Car accidents and falls are the main causes of nerve and brain injuries that can lead to permanent damage and dysfunction of a limb [1]. Furthermore, there are greater chances that this injury may develop anxiety in later stages [2]. Consequently, various kinds of the neuroprotective agent are tested preclinically and clinically on different steps of cascade in neuronal damage to ameliorate the aforementioned issues, thereby restoring their function by preventing further damage [3]. One of the most pertinent neuroprotective drugs is Dexmedetomidine (DEXM) which was approved in 1999 by the FDA as a short term (<24 h) sedative agent [4,5]. In 2008, the USA granted another indication of DEXM for use as a sedative agent in non-tracheal intubated patients before or during surgery [6,7,8].

Accordingly, the European Union approved DEXM in 2011 as an adult sedative agent for ICU [9]. A selective α2 agonist, DEXM has broad-spectrum effects, including being analgesic, sedative, and anxiolytic, with minimal depression of pulmonary function [9]. Previous works illustrated that DEXM has proved a crucial tool for critical patients by minimizing ventilation free hours [8,10]. On the other hand, it also attenuates delirium, improves postoperative cognitive dysfunction, and acts as a sympatholytic agent [11,12,13]. Recently, it caught the attention of researchers and clinicians due to its cardioprotective [14], renoprotective [15], pulmonoprotective [16], and neuroprotective effects [17]. However, there has been very limited research regarding DEXM as a neuroprotective agent against neuronal injury and only a few review articles have been reported. To the authors’ best knowledge, there is not a single review article on the neuroprotective effect of DEXM against neuronal injury. Therefore, to bridge this literature gap, an attempt has been made to provide the latest updates on the most recent contributions about DEXM, and which demonstrate the various growing research directions on its neuroprotective effects. The main contributions of this work are:It reviews recent studies in various animal models depicting DEXM’s anti-inflammatory role.Moreover, it elaborates the longitudinal studies of pre-and post-clinical trials on DEXM, which prove its ability to modulate apoptotic and necrotic events and also illustrate DEXM’s role in astroglia.Finally, the proposed study describes all the viable neuroprotective mechanisms of DEXM against nerve injury, which can support scientists and doctors in future studies.

The purpose of this article is to summarize and critically review the published data on the neuroprotective mechanisms of DEXM. This article also addresses the recent clinical applications of DEXM to have surfaced.

The rest of the paper is organized as follows. Section 2 illustrates DEXM’s pharmacology, while Section 3 presents DEXM’s neuroprotective mechanisms, and Section 4 concludes the study.

## 2. Pharmacology of Dexmedetomidine

### 2.1. Pharmacokinetics of Dexmedetomidine

DEXM is FDA approved IV drug that extensively undergoes a first-pass effect with a bioavailability of 16% [18]. It shows a better intranasal sedative and anxiolytic effect in comparison to clonidine [19]. DEXM exclusively binds with plasma protein (albumin) has the ability to cross the placenta and Blood-Brain Barriers (BBB) but its teratogenicity effect has not yet been fully explored [20].

DEXM is mainly metabolized in the liver via N-glucuronidation through uridine 5′-diphospho-glucuronosyltransferase (UGT2B10, UGT1A4) and cytochrome P450 (CYP2A6). The average half-life of DEXM in a healthy individual is 2.1–3.1 h, while in ICU it is 2.2–3.7 h.

### 2.2. Pharmacodynamics of Dexmedetomidine

The pharmacodynamic effect of DEXM is mainly dose-dependent. The biphasic hemodynamic responses, such as hypotension or hypertension, produce low or high plasma concentrations, respectively [21]. 

The concentration-dependent hypnotic and sedative action of DEXM is mediated through the activation of the α2 adrenergic receptor, which is anatomically located in the locus coeruleus [22]. The peculiar and unique pharmacodynamic properties of DEXM ensure “cooperative sedation”, in which even patients in the asleep stage can be easily aroused [23]. DEXM also can reduce pain via the α2 adrenergic receptor that alters perception, weakens the anxiolytic effect, and decreases the post-operative opioid need [24].

### 2.3. Adverse Effects of DEXM

DEXM’s main adverse effect is its hemodynamic instability because of α2 adrenoreceptor receptor activation that causes hypertension due to vasoconstriction, which leads to reflex bradycardia via carotid or aortic baroreceptor mediated autonomic activation. Moreover, DEXM is dose-dependent: a high dose of 1 or 2 μg/kg of DEXM over 2 min is required for rapid sedation. However, at the aforementioned dose concentration, an irregular and obstructive pattern of respiration has been reported [20].

The incidence of ventricular tachyarrhythmia, including ventricular fibrillation and ventricular tachycardia, is reduced after DEXM infusion during surgery. Conversely, DEXM shows no effects on atrial fibrillation [25], but overdose may cause I or II-degree atrioventricular blockage [20]. 

Newborn piglets experienced hypoxia and therapeutic hypothermia at a loading dose of 1 μg/kg with a maintenance infusion of 1.0 to 0.6 μg/kg/h, due to underdevelopment of the neonate cytochrome P450 enzyme and glucuronidation system. A few of the more adverse effects of DEXM include nausea, vomiting, and xerostomia [26].

### 2.4. Dexmedetomidine a Selective Alpha 2-Adrenoceptor Agonist

In 1948, Ahlquist challenged the opinion that adrenergic receptors were excitatory or inhibitory by differentiating the adrenergic receptors into an alpha and beta subtype [27]. The first α2-adrenoceptor agonist was manufactured in the early 1960s as a nasal decongestant (clonidine) [28]. Paton and his co-workers in 1996 found that a subclass of alpha adrenoceptors, located presynaptically, regulate the release of neurotransmitters [29]. This led to the subdivision of alpha adrenoceptors into postsynaptic alpha-1 and presynaptic alpha-2 adrenoreceptors. Other members of the α2 agonist receptors with α2/α1 selectivity include DEXM (1620:1), xylazine (160:1), Detomidine (260:1), Medotomidine (1620:1), clonidine (220:1), and Romifidine (affinity has not been confirmed yet) [30]. Alpha 2 adrenergic receptor is a transmembrane receptor that was differentiated into three iso-receptors, α2a, α2b, and α2c by Bylund. In contrast, its fourth type is only found in humans and is known as the α2d receptor, which is shown in Table 1. α2 receptor mediates effects through the potentiation of guanine-nucleotide regulatory binding proteins (G inhibitory protein), which ultimately attenuate the cAMP level. α2-adrenoceptor is widely distributed in the body, including the kidneys, pancreas, blood vessels, platelets, and eyes. Nonetheless, DEXM exerts its effects on α2-adrenoceptor distributed areas [23]. All subtypes of α-2 adrenoceptor perform different functions that include sedation, hypnosis, analgesia, sympatholysis, and neuroprotection via α-2A subtype [31,32]. Contrarily, the α-2B subtype is responsible for the suppression of central shivering, which causes peripheral vasoconstriction and analgesic effects on different sides of the spinal cord [33]. On the other hand, the stimulant induces locomotor activity, adrenaline outflow from the adrenal medulla, and mediation of cognitive sensory processing and emotional behavior, such as mood being controlled by α-2C [32]. In addition, DEXM performs all functions by binding with α-2 adrenoceptor [34,35], which links with the heterotrimeric transmembrane Gi protein that inhibits adenylyl cyclase. Inhibition of adenylyl cyclase decreases the formation of 3′5′cyclic adenosine monophosphate (cAMP), which acts as a second messenger. Gi protein activation opens inward potassium K^+^ rectifying channels that result in neuronal cell membrane hyperpolarization, which ultimately decreases the excitability of neuronal cells. On the other hand, the α-2 receptor also functions on the Go protein that inhibits Ca^2+^ entry into the cell and inhibits phospholipase C activity. On the other hand, phospholipase C inhibition attenuates Protein Kinase C (PKC), which has many roles in oxidative stress and apoptosis. Additional randomized clinical data also reported the binding of α-2 adrenoceptor with another undetermined G protein that works on sodium hydrogen (Na^+^/H^+^) exchanger [34,35], which is summarized in Figure 1.

### 2.5. Dexmedetomidine and Imidazoline Receptor

In 1987, Ernsberger et al. were the first to discover imidazoline binding sites in the ventrolateral medulla [41], belonging to a non-adrenergic family identified by compounds with an imidazoline moiety [42]. There are three imidazoline receptor subtypes found (I1, I2, and I3), among them, I2 is found in the medulla and brain, mainly the frontal cortex [36]. The neuroprotective effect of DEXM is mainly modulated by I2 noradrenergic receptors, not by I1 or I3 noradrenergic receptors [42]. I2 receptors associate with several distinct proteins but the identities of these proteins remain elusive. Its heterogeneous structure includes I2A and I2B subtypes, a division based on binding ability to the amiloride drug. Some data reported the evidence of DEXM binding with I2, causing Ca^2+^ influx into chromaffin cells and protecting neurons from damage. Nonetheless, DEXM’s neuroprotective effect via I2 requires further investigation, as few of its molecular mechanisms do not provide implicit information [43].

Manami Ingaki et al. [44] reported that Thapsigargin (non-competitive inhibitor of the Sarco/endoplasmic reticulum Ca^2+^ ATPase) can induce apoptosis and increase the influx of Ca^2+^. DEXM suppresses the Ca^2+^ level via the I2 receptor present on the outer membrane of mitochondria. The long-term potential (LTP) (a process of signal transmission in excitatory neurons of the hippocampus) is believed to be involved in the construction of neuronal circuits during learning and memory. DEXM can reduce LTP [45], and Yohimbine (α2 adrenoreceptor antagonist) is unable to fully abolish the DEXM effect on LTP, but combining with BU 224 hydrochloride (an imidazoline type 2 receptor antagonist) can reverse its effect. This information proves that DEXM also can bind with the imidazoline type 2 (I2) receptor [43,45].

## 3. Evidence of Dexmedetomidine as a Neuroprotective Agent

The role of DEXM as a neuroprotectant agent has been confined to animal experimental models. Even in laboratory animal models, the underlined neuroprotective mechanisms are not explicitly understood. Various DEXM neuroprotective properties are briefly discussed below.

### 3.1. Role of Dexmedetomidine in Overview Pathogenesis of Neuronal Damage and Death

Neural damage pathogenesis is a complex process, as it involves multiple molecular pathways. Numerous neuroprotective agents are applied to different types of neuronal injury such as necrosis, apoptosis, oxidative stress, and so on. DEXM exhibits a neuroprotective ability by modulating inflammatory markers and molecular pathways, which are summarized below [43].

### 3.2. The Influence of Dexmedetomidine against Necrosis

Necrosis is a type of irreversible cell death that usually occurs in acute phase injury. The pathophysiology of irreversible cell death consists of many components including the pre-necrotic event (pyroptosis) and the release of inflammatory markers, excitotoxic neurotransmitters, reactive oxygen species, and so on. All these components play a vital role in the pathogenies of cell death [46]. The signal transmission begins with a wave of depolarization and then repolarization that causes action potential (AP). The AP process exchanges cations of sodium (Na^+^) with potassium (K^+^), and during this process, the neurotransmitter acts as a chemical messenger. It is released from pre-synapses and binds to the corresponding receptor on a postsynaptic membrane, which causes depolarization or hyperpolarization [47,48]. Neurotransmitters are released in response to many processes, such as signal transmission, the release of hormones and peptides, and cell injury. Previous data have demonstrated that excitatory neurotransmitters such as glutamate and aspartate are released from the nerve terminal in response to nerve damage. These excitatory neurotransmitters also precipitate further nerve damage [49,50]. Additionally, this causes dysfunction of Na^+^/K^+^ ATPase, which leads to membrane hyperpolarization. Imbalance in levels of neurotransmitters such as catecholamine (CAT) increases nerve sensitivity to glutamate. The glutamate release by chelation of extracellular Ca^2+^ ions is associated with vesicular transportation. Glutamate binds to the NMDA (N-methyl-D-aspartate) receptor, which is a subtype of the glutamatergic receptor [49]. Furthermore, it activates the nitric oxide signaling pathway and enhances the intracellular Ca^2+^ ions that subsequently activate intracellular catabolic and proteolytic enzymes, for instance, protease and lipase [51,52]. These enzymes generate oxygen-free radicals that damage the membrane and results in cell death. Glutamate secretion is also regulated by mitogen-activated/extracellular signal-regulated kinase (MEK) [53].

Experimental models corroborate DEXM’s ability to modulate glutamate, and its inhibitory effect releases glutamate by reducing the release of CAT. DEXM inhibits glutamate release by binding with the α2 receptors that are expressed on neurons. This reduces the CAT release along with the CAT-associated nerve sensitivity for glutamate [53,54]. Additionally, DEXM directly blocks the MEK pathway and voltage-dependent calcium channel. It improves the expression of excitatory amino acid transporter 1 by increasing the N-methyl-D-aspartate receptor (NMDA) that decreases glutamate [55,56]. Excitatory Amino Acid Transporter 3 (EAAT3) is a member of the glutamate transporter family that removes glutamate from the synaptic cleft and extrasynaptic sites via glutamate re-uptake into glial cells and neurons. DEXM enhances EAAT3 expression [54,55]. The aforementioned discussion concludes that DEXM could prevent neuronal damage by inhibiting neurotransmitter release, as depicted in Figure 2.

Pyroptosis is a proinflammatory form of cell death that regulates necrosis. The clinical features of pyroptosis include rapid cell membrane rapture, cellular swelling, DNA fragmentation, and extravagation of cellular components resulting in the death of a neighboring healthy or normal cell, and inducing necrosis. The inflammasome is a protein that works as a sensor to detect cellular damage and invading pathogens. The LPS is released in response to pyroptosis. The LPS upregulates leucine rich repeated kinase (LRRK), nucleotide-binding domain (NLRP3), and apoptosis-associated speck like protein, Gasdermin D expression. The assembly of NLRP3 leads to the release of caspase 1 depending on cytokines IL-1β, IL-18 as well as Gasdermin D which mediate pyroptosis [57,58]. Previous data reports that cells treated with DEXM were found to protect against LPS inducing pyroptosis. Histone is a nuclear protein released from a dying cell into the extracellular space. LPS challenge the cell increase concentration of histone; it has been found that DEXM markedly inhibits histone release after LPS challenge [53,58,59].

### 3.3. The Response of Dexmedetomidine against Apoptosis

Apoptosis (programmed cell death) usually occurs in a late phase of injury and has an integral role in the normal development of a tissue or organ. The process of apoptosis involves B cell lymphoma/leukemia-2 (Bcl-2), a family of pro and anti-apoptotic regulatory proteins. The process of cellular death is delicately balanced by these proteins, which assess the integrity of the mitochondrial membrane, the release of cytochrome c, and potentiators of other apoptotic factors. BCL Associated Death protein (BAD) is a pro-apoptotic protein that is provoked by protooncogene protein C-akt (AKT) through phosphorylation of its serine residue. After AKT activation, BAD attaches to a cytosolic protein called 14-3-3 to reduce BCL-XL (anti-apoptotic protein), which delicately inhibits programmed cell death by binding with Bax (pro-apoptotic protein) [59].

It can be inferred that BCL-2 and BCL-XL proteins perform their anti-apoptotic effects by preventing the release of cytochrome C from mitochondria. Moreover, this also prevents Bax translocation and maintains mitochondrial membrane potential. DEXM potentially exhibits an anti-apoptotic property by regulating many pro and anti-apoptotic proteins, as it increases BCL2 [59], pERK concentration, and Murine Double Minute 2 (MDM-2). An increase in MDM-2 results in a reduction of pro-apoptotic mediator p53 [60]. Subsequently, it also increases Focal Adhesion Kinase (FAK), as well as the concentration of phosphorylated extracellular signal regulator kinase 1/2 (ERK1/2) [61]. FAK regulates its anti-apoptotic property by activating Phospohoinositide 3′-OH Kinase (PI3K) and the AKT pathway [62]. FAK simultaneously activates Nuclear Factor-kB (NF-kB), Inhibitor of Apoptosis Proteins (IAP), and IAP subtypes such as cIAP-1, cIAp-2, and XIAP, which alleviates apoptosis by inhibiting the caspase 3 activation [43].

The global cerebral ischemic insult may activate adaptive pathways and alter gene expression within the area of injury. Hypoxia Inducible Factor (HIF-1α) is expressed in response to hypoxia and it exponentially escalates several other genes, such as Vascular Endothelial Growth Factor (VEGF), erythropoietin, and DNA damage response 1 (REDD1/RTP801). The role of REDD1 is controversial in injury, but the reported data give evidence that it suppresses lactate dehydrogenase LHD release from neurons [58]. Data also indicate that DEXM upregulates HIF-1α, VEGF [63], and REDD1 [62] expression to protect the cell from oxygen-glucose deprivation induced injury. Oxygen glucose deprivation (OGD) insult is commonly used in vitro for hypoxia/ischemic modeling [64]. OGD exposure to cells causes deterioration of mitochondrial function and cell viability loss, and this death is usually by apoptosis (55%) and not by necrosis (only 7%) [63]. Furthermore, 1 µM of DEXM through the I2 imidazole receptor reduces the OGD induced cell death apoptosis to 22% and necrosis to 2% [62]. Moreover, a cell pretreated with BU 224 or idazoxan (I2 imidazoline antagonist) inhibits the DEXM protection by 82.3% and 92.4%, respectively. DEXM binds the I2 imidazole receptor, which activates the PI3K/AKT pathway and increases the HIF1α, VEGF, and REDD1 gene expression [62]. The imbalance between two pathways (phosphoinositide 3′-OH kinase (PI3K) and AKT protein kinase) plays a significant role in apoptosis [62], as these signaling pathways (PI3K/AKT) act with anti-apoptosis to enhance cell survival and proliferation [65]. DEXM phosphorylates AKT and increases expression of PI3K [65,66]. Furthermore, DEXM counteracts the OGD influenced inhibition of the p38 Mitogen Activated Protein Kinase (MAPK)/ERK1/2 pathway by elevating the phosphorylated level of p38 MAPK and ERK1/2 [67]. Caspase-3 expression was further investigated to observe the DEXM anti-apoptosis effect that protects against OGD-induced injury. Caspase-3 activation was increased by 8.7-fold due to OGD insult, while DEXM mitigated OGD induce caspase-3 expression to 1.8-fold [62]. The role of DEXM in apoptotic events is demonstrated in Figure 3. 

### 3.4. The Response of Dexmedetomidine against Immunologic Response

The significance of immunological or inflammatory response is not entirely clear because, on the one hand, it exacerbates oxidative stress, and on the other hand its phagocytes kill or damage nerve cells and help maintain neurogenesis. Inflammation or neuroinflammation is a complex process that involves many molecules, including pro-inflammatory cytokines (interleukins, TNF), chemokine, and inflammatory cells (monocytes, macrophages, Natural Killer Cell (NKC), astrocytes, and lymphocytes). Interleukin 1β or simple IL-1β has many names, such as a leukocytic endogenous mediator, mononuclear cell factor, leukocytic pyrogen, lymphocyte activating factor, and so on. It is a pro-inflammatory cytokine that is secreted by phagocytes (macrophage and monocytes). During acute insult, IL-1β is first released by microglia and then by astrocytes. Moreover, IL-1β has the ability to induce the release of other cytokines such as Tumor Necrotic Factor- α (TNF-α) and IL-6, which are released from astrocytes and microglial cells. IL-6 acts as a pro and anti-inflammatory cytokine, which affects metabolism, organ development, and hematopoiesis. In addition, IL-6 plays an essential role in infection, inflammation, neurodegenerative disease, and other ischemic injuries.

IL-1β induces mRNA expression of IL-6 in glioma cells through many pathways like phosphorylation of p38 Mitogen-Activated Protein Kinases (MAPK), Stress Activated Protein Kinase (SAPK)/c-Jun-N Terminal Kinase (JNK), and nuclear factor kappa-B kinase IκB/NF-κB [68,69]. IL-1β activities and synthesis IL-6 by binding within SMase/Src kinase/NMDA receptor through Calmodulin-Dependent Protein Kinase II and cAMP Response Element-Binding Protein (CamKII/CREB) activation pathway. DEXM exerts neuroprotective effects directly and indirectly by astrocytes as well as reduces the number of activated macrophages that involves in inflammation. Cell pretreated with DEXM decreases GFAP level (marker for astrocytes) [70] and CD116B level (marker for macrophage and monocytes) [70,71]. Previous work reported that DEXM alleviates IL-1 β-inducing-IL-6 release but it exerts very little effect on IL-6 suppression [72,73].

It has been experimentally proven in [72] that DEXM decreases IL-6 through suppression of IL-β but independently of the adenylyl cyclase-cAMP pathway, as 8-Bromo cAMP (analog of cAMP) increases the IL-6 level and Forskolin (direct activator of adenylyl cyclase) significantly magnifies the IL-1β-induced-release of IL-6. However, DEXM is unable to affect the forskolin-induced cAMP accumulation [72]. Yohimbine (α2-adrenoceptor antagonist) did not invert the suppressive effects of DEXM on the IL-1β-induced IL-6 release. It seems that DEXM, by employing α2-adrenoceptors, has a marginal chance of alleviating the IL-1β-evoked IL-6 release [72]. The above discussion signifies that more work needs to be done to accurately identify the molecular mechanisms.

DEXM not only reduces IL-β and IL-6 but also other pro-inflammatory cytokines such as TNF-α and NF-κB [74,75]. Meanwhile, the α2 receptor on the medulla oblongata is stimulated by DEXM, which alleviates the Sympathetic Nervous System (SNS) and dominant Parasympathetic Nervous System (PNS). Moreover, DEXM significantly decreases systemic cytokine levels in association with an enhancement in the discharge frequency of the cervical vagus nerve in response to endotoxin [70,76]. The inflammatory mediators are not only released by damaged or injured cells but also from supportive cells of the nervous system. Glial cells are the supportive cells of the nervous system that include oligodendrocytes, astrocytes, ependymal cells, and microglia. Glial cells not only help in neuronal repair but also support regeneration and maintain migration hemostasis and BBB. Astrocytes perform a complex function in both healthy and diseased cells of the nervous system.

On this basis, they are divided into good and bad astrocytes. Data studies reported that DEXM influences inhibitory effects by reducing astrocyte overactivation in response to nerve injury, or ischemic or reperfusion damage. Longitudinal studies also suggest that DEXM inhibits attenuate inflammatory mediator TNF-α and glial fibrillary acidic protein, which suggests DEXM has a neuroprotective effect [43]. DEXM and BDNF (Brain-Derived Neurotrophic Factor) provide a significant neuroprotective effect. DEXM enhances Bdnf4 and Bdnf5 transcription and also increases the BDNF in astrocytes through the extracellular signal-regulated kinase-dependent pathway, subsequently providing neuroprotection. As BDNF is a neurotrophin found in the healthy and diseased brain and body, evidence suggests that BNDF is associated with neurotransmitter regulation, neuronal plasticity, maintenance, and survival [77].

To summarize, DEXM treatment has been observed to eliminate or alleviate neuroinflammatory mediators (IL-6, TNF-α, NF-κB) and also increase Bdnf4 and Bdnf5 transcription and BDNF in astrocytes, which ultimately produces its neuroprotective effect, as summarized in Table 2.

### 3.5. Clinical Evidence of Dexmedetomidine as Neuroprotective Agent

Some randomized clinical trials that suggested DEXM as a neuroprotective agent are shown in Table 3. Some randomized clinical trials that suggested DEXM is a neuroprotective agent are also displayed in Table 2. Xiahong Luo et al. conducted a randomized clinical trial to explore the neuroprotective efficacy of DEXM. Sixty glioma patients underwent craniotomy resection, among them thirty patients received IV 1 μg/kg 10 min of DEXM with a maintenance dose of 0.4 μg/(kg/h). It was found that the DEXM could significantly reduce serum expression of an inflammatory mediator (TNF α, IL-6), inhibit free radical generation (superoxide dismutase), and stabilize hemodynamic parameters (Mean arterial pressure and heart rate) [19]. Xiahong Luo et al. [19] and Ashish Bindra et al. [78] ruled out the brain injury biomarkers: Neuron-Specific enolase of Enzyme (NSE) (enzyme release during neuronal injury) and S100b (express by type 2 astrocytes). These biomarkers cause cerebral insult in chronic temporal lobe epilepsy. Intraoperative DEXM treatment attenuates S100b and NSE, which suggests its neuroprotective property during epilepsy surgery [78].

Brain-Derived Neurotrophic Factor (BDNF) is a member of the neurotrophin family, which has an important role in axonal growth, synaptic plasticity, and neuronal survival. An elevated level of BDNF in the CNS can protect neurons from ischemic injury [79,80]. A randomized clinical trial suggested that IV DEXM infusion during surgery improves cognition after carotid endarterectomy by increasing BDNF concentration [71]. It helped by attenuating the duration of postoperative delirium [79]. The above studies provide a vital theoretical aspect for the application of DEXM in neuroprotection.

## 4. Conclusions

This work demonstrates the role of DEXM as a neuroprotective agent, which has been tested in various experimental models. It has been corroborated from the literature survey that DEXM abolishes its neuroprotective effects through the upregulation of α2 adrenoreceptor. DEXM imidazoline moiety binds with the I2 receptor, which is present on the outer surface of mitochondria that reduces Ca^2+^ modulated apoptosis. Moreover, preclinical data inferred that DEXM considerably decreases neuroinflammation and neurodegeneration following neurological insult. After a rigorous literature survey on the DEXM drug, the following decisive concluding remarks can be made:(i)Clinical investigations of DEXM should be designed to verify the efficacy of different steps of the cascade of neuronal damage.(ii)DEXM improves neuroinflammatory behavior by suppressing inflammatory mediators. It not only controls apoptotic signaling pathways but also decreases oxygen-free radical generation.(iii)The randomized clinical trials on DEXM suggest its ability to enhance BDNF to protect the brain from ischemic insult.(iv)DEXM not only improves cognition but also attenuates the duration of postoperative delirium.

The aforementioned conclusions regarding the neuroprotective agent (DEXM) can serve as a benchmark for researchers and doctors to substantiate DEXM’s efficacy in exerting anti-inflammatory properties that ameliorate and prevent nerve injuries. However, many areas are yet to be explored, for instance, certain extended applications of DEXM require further evaluation to ensure the safe use of DEXM before it is designed in a clinical trial. It is imperative to carefully select patients and to determine the appropriate dosage, as DEXM usage is still limited in the pediatric population.

## Figures and Tables

**Figure 1 brainsci-11-00846-f001:**
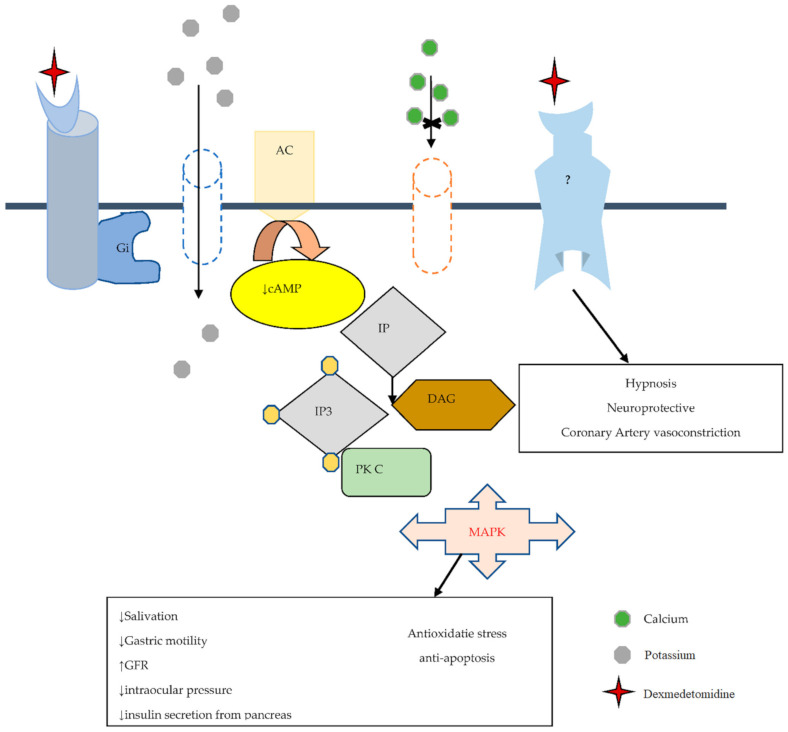
The possible DEXM neuroprotective mechanism is mediated by α2 adrenoreceptor, which links with heterotrimeric transmembrane G inhibitory protein (Gi), which inactivates Adenyl cyclase (AC). Inactivation of AC attenuates the cAMP level (act as the second messenger). Gi opens inward, rectifying the K+ channel and leading to neuronal membrane hyperpolarization. Normally phosphoinositide (PI)hydrolysis to produce hydrolysis produces inositol 1,4,5-trisphosphate (IP3) and diacylglycerol (DAG) and activates PKC, and has a role in oxidative stress and apoptosis. Agonist agent of α2 receptor (DEXM) via G0 blocks Ca^2+^ translocation and inhibits phospholipase C activity, reducing PKC activity. “↑” increase and “↓” decrease.

**Figure 2 brainsci-11-00846-f002:**
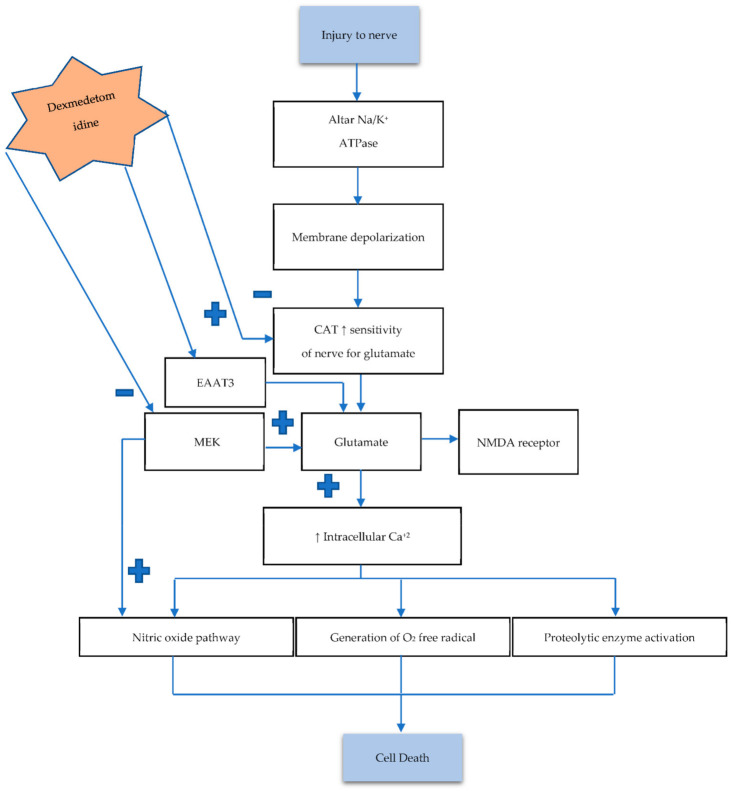
Nerve injury causes dysfunction of NA/K ATPase, leading to an imbalance between neurotransmitters. Increasing CAT level enhances the sensitivity of nerves to glutamate (Glu). Excessive Glu increases intracellular Ca^2+^, which in return activates the aforementioned pathways, resulting in cell death. Activation of the mitogen-activated/extracellular signal-regulated kinase (MAK) pathway also increases Glu concentration. The α2 adrenoreceptor exerts its possible neuroprotective mechanism by hyperpolarization, which blocks the Ca^2+^ entry into presynase. Low Ca^2+^ diminishes neurotransmitter release (CAT) and Glu and also decreases nerve sensitivity for Glu. DEXM also enhances Excitatory amino acid transporter 3 (EAAT3) activity, which removes Glu from the synaptic cleft. Hyperpolarization also reduces NMDA receptors, which also decreases firing and reduces intracellular Ca^2+^. DEXM also blocks MEK, resulting in decreased excitotoxic neuronal injury.

**Figure 3 brainsci-11-00846-f003:**
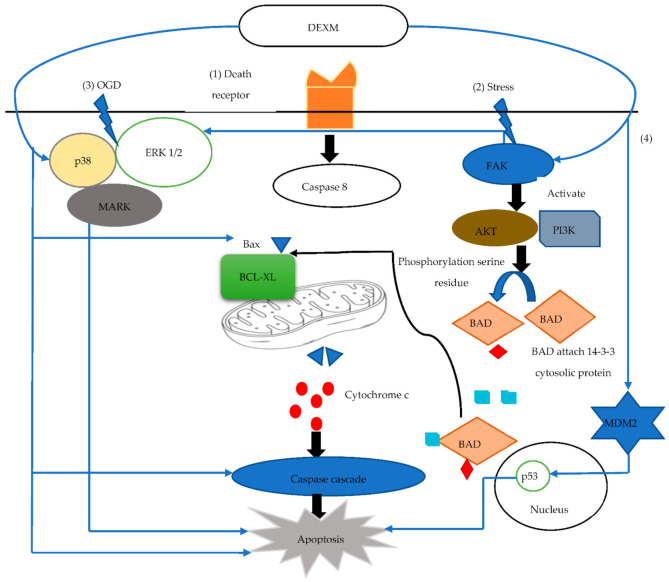
(1) Activation of death receptor release Caspase 8 that helps Ba bind with BCL-XL and releases cytochrome C (Cyt C) from mitochondria. Cyt C activates caspase cascade induced apoptosis. (2) Stress activates FAK, which activates AKT/PI3 K and ERK ½. AKT helps BAD serine residue phosphorylation. Phosphorylated BAD attaches cytosolic 14-3-3 protein that unbinds Ba with BCL-XL, attenuating apoptosis. (3) OGD induces necrosis by decreasing the p38 Mitogen-activated protein kinase (MAPK)/ERK1/2 pathways that attenuate apoptosis. (4) MDM2 reduces p53 activity, which induces apoptosis. DEXM enhances FAK activity and also phosphorylates AKT and increases expression of PI3K, increasing detachment of Ba-BCL-XL. DEXM counteracts OGD-influenced inhibition by elevating the phosphorylated level of p38 MAPK and ERK1/2. DEXM modulates p53 activity via Mdm2.

**Table 1 brainsci-11-00846-t001:** Alpha 2 receptor subtypes, anatomical location, and functions.

Alpha 2 Receptor Subtype	Anatomical Location	Functions	References
α2a	CNS including the brain, spinal cord, and locus coeruleus	Regulates the phase of awareness produces anesthetic, sympatholytic responses, helps in arousal, and vigilance	[36,37,38]
α2b	Liver, spleen, heart, Thalamus, smooth muscle cells of the blood vessel.	Vasoconstrictive.	[39,40]
α2c	Hippocampus, locus coeruleus, especially in basal ganglia, platelets.	Control anxiolytic and analgesic effects, hypnotic-sedative actions	[38,39,40]

**Table 2 brainsci-11-00846-t002:** Summary of DEXM-associated neuroprotective effects.

References	Mechanisms	Animal Model	Marker	DEXM Effect
[55]	Excitotoxicity	Neonatal rats randomly allocated to 4 groups (*n* = 36 each)	Glutamate	↑ EAAT3 expression, ↓ nerve sensitivity for glutamate.
[59]	Apoptosis	Adult male Sprague Dawley rats randomly allocated to 4 groups (*n* = 10 each)	Bcl2, FAK	Upregulates anti-apoptotic marker
[17]		Adult male C57BL/6 J mice *n* = 80	MDM2, p53 pathway	↓ p53 activity via Mdm2
[62]		Rat glioma C6 cells Cell culture in Dulbecco’s modified Eagle’s Medium (DMEM)	PI3k/AKT	Phosphorylates AKT, upregulates PI3k expression
[61]		Male Sprague-Dawley rats n = unknown	Caspase 3	Upregulates FAK that activates IAP that attenuate caspase 3
[72]	Inflammatory mediator	Rat C6 glioma cells. Cell culture in DMEM	IL-1ꞵ induces IL-6 release	↓ IL-1ꞵ induces IL-6 release independently adenyl cyclase cAMP pathway
[71]		Long–Evans female rats	Systemic cytokines TNF-α, NF-kB, IL-1ꞵ, IL-18	Significantly decreases systemic cytokines level
[58]	Stepsis	Human astrocyte 1321N1 cells and rat neuron PC12 cells. Cell culture in DMEM	LHD, NLRP3, ASC, Caspase 1 inducing IL-1ꞵ, IL-18	Inhibits NL RP3 inflammasome assembly
[43]	Astroglia	Astrocytes, Cell culture	Astrocytes	Decrease over activated astrocytes

Note: “↑” increase and “↓” decrease.

**Table 3 brainsci-11-00846-t003:** Randomized clinical trials of Dexmedetomidine as a neuroprotective agent.

Condition	Number of Cases	Outcome	References
Brain protection in patients undergoing craniotomy resection of glioma.	60 cases	DEXM stabilized hemodynamics, attenuated inflammation, and inhibited the generation of free radicals.	[19]
Cerebroprotection during epilepsy surgery.	19 cases	Treatment with DEXM low S100b level	[78]
Improves cognition after carotid endarterectomy.	49 cases	DEXM was neuroprotective in the stroke model by reducing TNF-α and IL-6 and enhancing BNDF.	[71]
Reduce postoperative delirium (POD) in elderly patients with mild cognitive impairment (MCI) after joint replacement surgery.	80 cases	DEXM treatment significantly improved POD MCI in elderly patients.	[79]
A double-blind, randomized, and placebo-controlled study.	40 cases	DEXM enhances plasma concentrations of BDNF caused by the anesthetic agent.	[80]

## Data Availability

Not applicable.

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
