# Peer review of "The Current Role of Dexmedetomidine as Neuroprotective Agent: An Updated Review"

_brainsci, 2021, doi:10.3390/brainsci11070846_

Round 1
Reviewer 1 Report
Numerous linguistic errors, for example:
line 85: it should be: DEXM has also
line 92: ... alpha2-adrenoceptor AGONIST was ...
The citations are poorly chosen, in the cited articles, there is no information that the authors refer to. Therefore, it needs to be verified.
line 55
unclear sentence missing the second part of the sentence
line 64-66
superfluous fragment in this part of the text, there should be a separate chapter about side effects, the adverse effects of dexmedetomidine include hypotension, hypertension, nausea, bradycardia, atrial fibrillation, and hypoxia. Overdose may cause a first-degree or second-degree atrioventricular block.
line 70-80
this is an excerpt from the authors' guidelines or comments from the previous reviewer, not an article text! Have the authors read their work?
line 88
in the quoted article [9] there is no information that DEXM binds to the imidazole receptor. In this sentence it says that it attaches to the alpha and I receptor. The next line says DEXM is selective, information contradicting. It was probably an article [29].
line 84
an incomprehensible sentence, is it about the half-life of the drug? The literature [for example 9] gives the t1/2 period of 2.2-3.7h or 2.1-3.1h.
line 96
the first invented drug, clonidine, is missing in the mentioned drugs, it is also worth comparing the selectivity of various drugs, for example: medetomidine (1620: 1), detomidine (260: 1), clonidine (220: 1), xylazine (160: 1)... Further, the authors mentioned that there are different families of receptors (alpha2a, alpha2b, alpha2c). However, they do not explain the differences between drugs from this group in their effects on these receptors and the consequences of these differences. The authors indicate different functions of the subtypes of these receptors; therefore such a combination would be valuable.
line 126
the authors write that DEXM is mainly modulated by I2, noradrenergic receptors, not by I1 or I3 noradrenergic receptors [29]. However, in the mentioned work ([29]), it is written:
dexmedetomidine (a a2-adrenergic agonist / I1 receptor agonist)
next:
dexmedetomidine had a postconditioning effect dependent on I1 receptors (Dahmani et al., 2010)
there is no information there about the action of DEXM only on the I2 receptor.
line 126-133
the nomenclature of the receptor families indicates α1, α2-adrenenoceptors and imidazoline I1, I2, I3-receptors. It is not clear why the authors use the name I1/2/3 noradrenergic receptors.
line 132
In the cited work [30] there is no information on the influence of DEXM via the I2 receptor.
line 196
The anion has a negative charge, K+ is a cation, anions are Cl- ions, for example.
The conclusions are too poor, mainly due to the clinical trials (chapter 3.5, table 2), which have not been analyzed in detail, which seems crucial in this kind of review.
Work requires serious improvements.
Author Response
Response to Reviewer 1 Comments
At first, we would like to thank you very much for the great efforts you spent on editing our paper. Also, thank you very much for giving us another chance to modify our paper based on the valuable comments given by the respected reviewers. Also, we would like to deeply thank the respected reviewers for their valuable comments to improve the clarity, the presentation and the language of our paper.
Secondly, we did our best in revising the paper based on all the valuable comments/concerns given by the reviewers as given in the revised manuscript and the reply letter.
Point 1: line 85: it should be: DEXM has also
Author response: Thank you for highlighting this mistake. The corresponding correction has been made.
Author action: We updated the manuscript by inserting the above changes in section 2, page 2 of the manuscript.
Point 2: line 92: ... alpha2-adrenoceptor AGONIST was
The foremost α2-adrenoceptor agonist was manufactured in the early 1960s as a nasal decongestant (clonidine)
Author response: Once again thank you for even correcting the small errors and proposing text corrections. Above text corrections have been corrected as per your suggestion.
Author action: We updated the manuscript by adding the above changes in section 2.3 and page 3
Point 3: The citations are poorly chosen, in the cited articles, there is no information that the authors refer to. Therefore, it needs to be verified
Author response: Thank you for valuable suggestion. The whole article has been carefully revised and verified by inserting relevant citations of given information. For instance, line 127-128 information has been provided in Reference No. 29 (in History section).29. Afonso, J.; Reis, F. Dexmedetomidine: current role in anesthesia and intensive care. Brazilian Journal of Anesthesiology 2012, 62, 118-133.Furthermore, more recent relevant citations have been included in the manuscript.
Author action: We updated the manuscript by adding relevant citation of line 127-128 and also carefully verifying the rest of citations.
Point 4: line 55 unclear sentence missing the second part of the sentence
Author response: Sorry for this silly typo, the above line should not have been there and it has been omitted.
Author action: We updated the manuscript by omitting the above line.
Point 5: line 64-66 superfluous fragment in this part of the text, there should be a separate chapter about side effects, the adverse effects of dexmedetomidine include hypotension, hypertension, nausea, bradycardia, atrial fibrillation, and hypoxia. Overdose may cause a first-degree or second-degree atrioventricular block
Author response: Yes, you are right, it would be more appropriate to add separate chapter for side effects which as per your instruction we have added as shown below.
DEXM main adverse effect is hemodynamic instability because of α2 adrenoreceptor receptor activation that causes hypertension due to vasoconstriction which leads to re-flex bradycardia via carotid or aortic baroreceptor mediated autonomic activation. Moreover, DEXM is dose-dependent: a high dose of 1 or 2 μg/kg of DEXM over 2 minutes is required for rapid sedation. However, at the aforementioned dose concentration, the irregular and obstructive pattern of respiration has been reported [19].
The incidence of ventricular tachyarrhythmia includes ventricular fibrillation and ventricular tachycardia is reduced after DEXM infusion during surgery. Conversely, DEXM shows no effect on atrial fibrillation [24] but overdose may cause ⅠorⅡdegree atrioventricular block [19].
New born piglet experiences hypoxia and therapeutic hypothermia at a loading dose of 1μg/kg with a maintenance infusion of 1.0 to 0.6μg/kg/h due to underdevelopment of neonate cytochrome P450 enzyme and glucuronidation system. A few of the more ad-verse effects of DEXM include nausea, vomiting, xerostomia [25].
Author action: We updated the manuscript by adding adverse effects of DEXM on page 3.
Point 6: line 70-80
this is an excerpt from the authors' guidelines or comments from the previous reviewer, not an article text! Have the authors read their work?
Author response: Thank you so much for pointing out this silly error. Actually, it was MDPI general instructions which we forgot to omit while writing our manuscript. However, from line 70-72, it is the layout of the rest of article while from 73-80, these are redundant lines which have been omitted now.
Author action: We updated the manuscript by omitting the above redundant lines (73-80).
Point 7: line 88
in the quoted article [9] there is no information that DEXM binds to the imidazole receptor. In this sentence it says that it attaches to the alpha and I receptor. The next line says DEXM is selective, information contradicting. It was probably an article [29].
Author response: Thank you for your valuable input. Actually, this is not the contradicting information as DEXM has the ability to bind with both alpha-adrenergic receptor and imidazole receptor and this information can be validated in references 45 and 46 and these pertinent citations now have been incorporated in the article
Author action: We updated the manuscript by adding the pertinent citations of above information on page 4.
Point 8: line 84
an incomprehensible sentence, is it about the half-life of the drug? The literature [for example 9] gives the t1/2 period of 2.2-3.7h or 2.1-3.1h.
Author response: Thank you for pointing out. The corresponding suggestion has been made.
Author action: We updated the manuscript by inserting the above-mentioned writing pattern of half-life on page 2.
Point 9: line 96
the first invented drug, clonidine, is missing in the mentioned drugs, it is also worth comparing the selectivity of various drugs, for example: medetomidine (1620: 1), detomidine (260: 1), clonidine (220: 1), xylazine (160: 1)... Further, the authors mentioned that there are different families of receptors (alpha2a, alpha2b, alpha2c). However, they do not explain the differences between drugs from this group in their effects on these receptors and the consequences of these differences. The authors indicate different functions of the subtypes of these receptors; therefore, such a combination would be valuable.
Author response: Yes, you are right such combination would be very helpful and as per your instruction, we added the α1/α2 selectivity of drugs and also mentioned the difference among alpha family members in tabular form as shown below:
Table 1. different type of alpha 2 receptor their anatomical location and function
Alpha receptor subtype |
Anatomical location |
Function |
References |
α2a |
CNS including brain, spinal cord and locus coeruleus |
regulates the phase of awareness, produces anesthetic, sympatholytic responses, help in arousal, and vigilance |
[37-39] |
α2b |
liver, spleen, heart, Thalamus, smooth muscle cells of blood vessel. |
Vasoconstrictive. |
[40,41]
|
α2c |
Hippocampus, Locus coeruleus, especially in basal ganglia, platelets. |
Control anxiolytic and analgesic effects, hypnotic-sedative actions |
[39-41] |
Author action: We updated the manuscript by adding above information on page 4.
Point 10 line 126
the authors write that DEXM is mainly modulated by I2, noradrenergic receptors, not by I1 or I3 noradrenergic receptors [29]. However, in the mentioned work ([29]), it is written:
dexmedetomidine (a a2-adrenergic agonist / I1 receptor agonist)
next:
dexmedetomidine had a postconditioning effect dependent on I1 receptors (Dahmani et al., 2010)
there is no information there about the action of DEXM only on the I2 receptor.
Author response: Yes, you are right that reference 29 mentioned the affinity of DEXM with all imidazoline receptors as well as alpha 2 adrenergic receptor but the neuroprotective effect of DEXM is mainly operated via I2 receptor.
Manami Ingaki et al [45] reported that Thapsigargin (non-competitive inhibitor of the Sarco/endoplasmic reticulum Ca2+ ATPase) can induce apoptosis and increase the influx of Ca+2. DEXM suppresses the Ca+2 level via the I2 receptor present on the outer membrane of mitochondria. The Long-Term Potential (LTP) (a process of signal transmission in excitatory neurons of the hippocampus) is believed to involve in the construction of neuronal circuits during learning and memory. DEXM has the ability to reduce LTP [46], Yohimbine (α2 adrenoreceptors antagonist) is unable to fully abolish the DEXM effect on LTP but combine with BU 224 hydrochloride (an imidazoline type 2 receptor antagonist) can reverse its effect. This information proves that DEXM also has the ability to bind with the imidazoline type 2 (I2) receptor [44,46].
The above information with correct citations has been added in the manuscript.
Author action: We updated the manuscript by adding above information with its correct citation on page 4.
Point 11: line 126-133
the nomenclature of the receptor families indicates α1, α2-adrenenoceptors and imidazoline I1, I2, I3-receptors. It is not clear why the authors use the name I1/2/3 noradrenergic receptors.
Author response: Actually, imidazoline receptors are also known as noradrenergic or non-adrenergic receptors. This term has been used in other papers as well. For instance, Ref [29] of manuscript also used the term ‘noradrenergic’
Author action: The reviewer concern has been replied here.
Point 12: line 132
In the cited work [30] there is no information on the influence of DEXM via the I2 receptor.
Author response: Influence of DEXM via the I2 receptor has been provided in Ref [30] (now becomes Ref [29]), section 2 under the heading “Effects at other sites”
Author action: The reviewer concern has been addressed here.
Point 13: line 196
The anion has a negative charge, K+ is a cation, anions are Cl- ions, for example.
Author response: Yes, you are right, the corresponding correction has been made.
Author action: We updated the manuscript by incorporating above correction on page 6, line 249.
Point 14: The conclusions are too poor, mainly due to the clinical trials (chapter 3.5, table 2), which have not been analyzed in detail, which seems crucial in this kind of review.
Author response: Thank you for valuable suggestion. As per your instruction Table 2 has been elaborated in the manuscript and also part of conclusion has been revised.
Some randomized clinical trials suggested that DEXM is a neuroprotective agent that has been discussed in Table 2. Xiahong Luo et al. conducted a randomized clinical trial to explore the neuroprotective efficacy of DEXM. Sixty glioma’s patients underwent craniotomy resection, among them thirty patients received IV 1μg/kg 10min of DEXM with a maintenance dose of 0.4μg/(kg/h). It found that the DEXM can significantly re-duce serum expression of an inflammatory mediator (TNF α, IL-6), inhibit free radical generation (superoxide dismutase) and can stabilize hemodynamic parameters (Mean arterial pressure and heart rate) [18]. Xiahong Luo et al [18] and Ashish Bindra et al [79] rule out the brain injury biomarkers: Neuron-Specific enolase of Enzyme (NSE) (enzyme release during neuronal injury) and S100b (express by type 2 astrocytes). These biomarkers cause cerebral insult in chronic temporal lobe epilepsy. Intraoperative DEXM treatment attenuates S100b and NSE that suggest its neuroprotective property during epilepsy surgery [79].
Brain-Derived Neurotrophic Factor (BDNF) is a member of the neurotrophin family which has an important role in axonal growth, synaptic plasticity and neuronal survival. An elevated level of BDNF in the CNS can protect neurons from ischemic in-jury [80,81]. A randomized clinical trial suggests that IV DEXM infusion during surgery improves cognition after carotid endarterectomy by increasing BDNF concentration [72]. It helps in attenuating the duration of postoperative delirium [80]. The above studies provide a vital theoretical aspect for the application of DEXM in neuroprotection.
Author action: We updated the manuscript by adding the detail of Table 3 on page 11 and adding more information in conclusion on page 12.
Yours Sincerely,
Dr. Zaara Liaquat et al.
Reviewer 2 Report
The manuscript reports information about current role of dexmedetmodine as a potential neuroprotective agent and its promising efficacy in clinical use. Authors discuss mainly pharmacodynamics of dexmedetomidine and based on these mechanisms, they explain its potential neuroprotective effect.
Major issue:
The part called “Pharmacodynamics and Pharmacokinetics of Dexmedetomidine” should be divided. The pharmacokinetics should be discussed separately and more detailed. The term “fast pass reaction” should be replace by “first pass effect”.
Author Response
Response to Reviewer 2 CommentsAt first, we would like to thank you very much for the great efforts you spent on editing our paper. Also, thank you very much for giving us another chance to modify our paper based on the valuable comments given by the respected reviewers. Also, we would like to deeply thank the respected reviewers for their valuable comments to improve the clarity, the presentation and the language of our paper.
Secondly, we did our best in revising the paper based on all the valuable comments/concerns given by the reviewers as given in the revised manuscript and the reply letter.
Point 1: The part called “Pharmacodynamics and Pharmacokinetics of Dexmedetomidine” should be divided. The pharmacokinetics should be discussed separately and more detailed. The term “fast pass reaction” should be replace by “first pass effect”.
Author Response: Thank you very much for your valuable suggestion. As per your instruction, “fast pass reaction” has been replaced with “first pass effect”. Moreover, Pharmacodynamics and Pharmacokinetics of Dexmedetomidine has been divided in two sections and also more detail has been provided in the manuscript and this detail is also shown below:
2.1 Pharmacokinetics of Dexmedetomidine
DEXM is FDA approved IV drug that extensively undergoes a fast pass effect with a bioavailability of 16%. It shows a better intranasal sedative and anxiolytic effect in comparison to clonidine [18]. DEXM exclusively bind with plasma protein (albumin) has the ability to cross placenta and Blood-Brain Barriers (BBB) but its teratogenicity effect has not yet fully explored [19].
DEXM is mainly metabolized in the liver via N-glucuronidation through uridine 5′-diphospho-glucuronosyltransferase (UGT2B10, UGT1A4) and cytochrome P450 (CYP2A6). The average half-life of DEXM in healthy individual is 2.1-3.1h while in ICU it is 2.2-3.7h.
2.2 Pharmacodynamics of Dexmedetomidine
The pharmacodynamic effect of DEXM is mainly dose-dependent. The biphasic hemo-dynamic response such as hypotension or hypertension produces low or high plasma concentration respectively [20].
The concentration dependent hypnotic and sedative action of DEXM is mediated through the activation of the α2 adrenergic receptor which anatomically located in locus coeruleus [21]. The peculiar and unique pharmacodynamic property of DEXM endures “cooperative sedation” in which patients even in the asleep stage can be easily aroused [22]. DEXM also has the ability to reduce pain via α2 adrenergic receptor that altered perception and weaken anxiolytic effect and decrease the post-operative opioid need [23].
2.3 Adverse Effect of DEXM
DEXM main adverse effect is hemodynamic instability because of α2 adrenoreceptor receptor activation that causes hypertension due to vasoconstriction which leads to re-flex bradycardia via carotid or aortic baroreceptor mediated autonomic activation. Moreover, DEXM is dose-dependent: a high dose of 1 or 2 μg/kg of DEXM over 2 minutes is required for rapid sedation. However, at the aforementioned dose concentration, the irregular and obstructive pattern of respiration has been reported [19].
The incidence of ventricular tachyarrhythmia includes ventricular fibrillation and ventricular tachycardia is reduced after DEXM infusion during surgery. Conversely, DEXM shows no effect on atrial fibrillation[24] but overdose may cause ⅠorⅡdegree atrioventricular block [19].
Newborn piglet experiences hypoxia and therapeutic hypothermia at a loading dose of 1μg/kg with a maintenance infusion of 1.0 to 0.6μg/kg/h due to underdevelopment of neonate cytochrome P450 enzyme and glucuronidation system. A few of the more ad-verse effects of DEXM include nausea, vomiting, xerostomia [25].
Author action: We updated the manuscript by inserting the above changes in section 2, page 2 of the manuscript.
Yours Sincerely,
Dr. Zaara Liaquat et al.
Round 2
Reviewer 1 Report
The corrections made by the authors seem to be sufficient.
The work is suitable for publication.
Reviewer 2 Report
The revised version of the manuscript is sufficiently amended. Authors appended the part describing the pharmacokinetic of dexmedetomidine with more details and provided adequate insight into this topic. Nevertheless, I did not find the “first pass effect” term correction.